

# ClpP regulates breast cancer cell proliferation, invasion and apoptosis by modulating the Src/PI3K/Akt signaling pathway

Juan Luo[1], Beilei Zeng[1], Chunfang Tao[1], Mengqi Lu[1] and Guosheng Ren[1,2]

[1] Chongqing Key Laboratory of Molecular Oncology and Epigenetics, The First Affiliated Hospital of Chongqing Medical University, Chongqing, China

[2] Department of Endocrine and Breast Surgery, The First Affiliated Hospital of Chongqing Medical University, Chongqing, China

## ABSTRACT

**Background**. Caseinolytic protease P (ClpP), which is located on the inner mitochondrial membrane, degrades mitochondrial proteins damaged by oxidative stress. The role of ClpP varies among tumor types. However, the expression pattern and biological functions of ClpP in breast cancer (BC) have not yet been investigated.

**Methods**. The Cancer Genome Atlas (TCGA) and Kaplan Meier-plotter database were used to analyze the expression level of ClpP in BC tissues, relationships with clinicopathological characteristics, and the influence on the prognosis of BC. Protein and mRNA expression levels of ClpP in BC cell lines and tissues were detected by quantitative real-time PCR, western blot and immunohistochemical (IHC) analyses. The colony formation assay, transwell assay and flow cytometric analysis were performed to assess various functions of ClpP. Western blot analysis was also conducted to determine the mechanism of ClpP.

**Results**. ClpP expression was markedly increased in BC cells and tissues. High expression of ClpP was significantly correlated with the T stage, estrogen receptor (ER) expression, and poor recurrence-free survival (RFS) in TCGA and Kaplan Meier-plotter database. ClpP silencing significantly inhibited proliferation, migration, invasion, and promoted apoptosis of BC cells, which resulted in suppression of the Src/PI3K/Akt signaling pathway. The gain-of-function assay confirmed partial these results.

# INTRODUCTION

Breast cancer (BC) is the most common malignant tumor among women worldwide (*Bray et al., 2018*). At present, the main treatment strategy for BC is surgical resection of the breast and axillary lymph nodes, in combination with adjuvant therapies, which include chemotherapy, radiotherapy, endocrine and targeted therapy (*National Comprehensive Cancer Network, 2019*). The rapid development of deep sequencing and molecular

Corresponding author
Guosheng Ren, rengs726@126.com

biotechnologies have allowed for the identification of new cancer targets and targeted therapeutic drugs.

The mitochondria have been a focus of cancer research since the 1950s, when it was discovered that cancer cells produce adenosine triphosphate (ATP) and employ different mechanisms to support cell growth than the normal surrounding tissues. Therefore, it was suggested that a defect in the mitochondrial mechanism will not only lead to increased glycolysis, but also to the transformation of normal cells into cancer cells (*Warburg, 1956*). The maintenance of mitochondrial function requires strict control of protein homeostasis via independent mechanisms for protein synthesis and degradation. In addition to the cytoplasmic ubiquitin/proteasome and protein quality control systems, the mitochondria of mammalian cells have three ATP-dependent protease families: Lon (*Wang et al., 1993*), FtsH (*Banfi et al., 1999*; *Casari et al., 1998*) and ClpXP (*Corydon et al., 1998*; *Kang et al., 2002*). These proteases regulate protein degradation and maintain protein quality control (*Goldberg, 2003*; *Sauer et al., 2004*). There is also evidence that an elevation to the proteostatic threshold can lead to the onset of various diseases, especially cancer, and maintain the stability of mitochondrial proteins in tumor cells (*Seo et al., 2016*).

The ClpXP protease complex is composed of two proteins: hexamers of a AAA+ ATPase (ClpX) and the tetradecameric peptidase caseinolytic protease P (ClpP) (*Bross et al., 1995*; *Corydon et al., 1998*; *Kang et al., 2002*), which contribute to the pathogenesis of human disease (*Gispert et al., 2013*). A recent study reported that ClpXP is upregulated in primary and metastatic human tumors, necessary to support tumor cell proliferation, motility and heightened metastatic competence *in vivo*, and correlated with shortened survival. However, the ClpP and ClpX subunits may not have completely overlapping function(s) in the tumor mitochondria (*Seo et al., 2016*).

ClpP is encoded by nuclear genes in mammalian cells and plays a central role in the quality control of mitochondrial proteins via the degradation of misfolded proteins. ClpP was first identified in bacteria and has since aroused interest as a potential anti-microbial therapeutic target (*Zeiler et al., 2012*). Studies have shown that bacterial ClpP inhibitors with a beta-lactone structure have antibacterial activities (*Bottcher & Sieber, 2008*; *Gersch et al., 2013*; *Szyk & Maurizi, 2006*). It has also been reported that mutations in human ClpP were associated with Perrault syndrome (*Jenkinson et al., 2013*). However, few studies have examined the role of ClpP in tumorigenesis.

The expression level of ClpP is greater in acute myeloid leukemia (AML) cells than in normal hematopoietic cells (*Cole et al., 2015*). Further research found that ClpP hyperactivation can lead to the death of leukemia and lymphoma cells due to selective proteolysis of mitochondrial proteome subsets involved in mitochondrial respiration and oxidative phosphorylation (*Ishizawa et al., 2019*). Cancer stem cells and chemo-resistant cells are highly dependent on oxidative phosphorylation (*Farge et al., 2017*; *Kuntz et al., 2017*; *Lagadinou et al., 2013*; *Marin-Valencia et al., 2012*; *Viale et al., 2014*). Hence, ClpP could be exploited as a novel target in cancer treatment. Therefore, the aim of the present study was to explore the expression pattern, biological functions and underlying mechanisms of ClpP as a novel target for the treatment of BC.

## MATERIALS & METHODS

### Tissue specimens

Human BC tissues and corresponding adjacent normal tissues were collected from patients who underwent surgery at the First Affiliated Hospital of Chongqing Medical University from 2014 to 2018, snap-frozen in liquid nitrogen, and then stored at $-80\,°C$. This study was approved by the Institutional Ethics Committees of the First Affiliated Hospital of Chongqing Medical University (approval no. 2019-208) and conducted in accordance with the tenets of the Declaration of Helsinki. Each participant signed an informed consent form prior to study inclusion.

### Cell culture

Seven human BC cell lines (T47D, MCF-7, ZR-75-1, SK-BR-3, MDA-MB-231, MDA-MB-468 and BT-549) and two normal mammary epithelial cell lines (MCF-10A and HBL-100) were obtained from the American Type Culture Collection (ATCC, Manassas, VA, USA). MCF-10A cells were cultured as described previously (*Debnath, Muthuswamy & Brugge, 2003*) and all other cell lines were cultured in Roswell Park Memorial Institute 1640 medium (Gibco BRL, Karlsruhe, Germany) supplemented with 10% fetal bovine serum (FBS) (Gibco BRL) at $37\,°C$ under a humidified atmosphere of 5% $CO_2$/95% air.

### Small interfering RNAs (siRNAs), plasmids, and transfection

ClpP-specific and the negative control siRNAs were synthesized by OriGene (OriGene Technologies Inc., Rockville, MD, USA). The following siRNAs sequences were generated: SR305388A-rGrCrUrCrArArGrArArGrCrArGrCrUrCrUrArUrArArCrATC; SR305388B-rGrUrUrUrGrGrCrArUrCrUrUrArGrArCrArArGrGrUrUrCTG; and SR305388C-rGrGrCrCrArUrCrUrArCrGrArCrArCrGrArUrGrCrArGrUAC. The expression vector pCMV6-Myc-DDK-ClpP was produced by OriGene (OriGene Technologies Inc., Rockville, MD, USA); the empty pCMV6-Myc-DDK vector was used as a control. Lipofectamine 2000 (Invitrogen Corporation, Carlsbad, CA, USA) was used for siRNAs and plasmids transfection, in accordance with the manufacturer's protocols.

### RNA extraction and quantitative real-time PCR (RT-qPCR)

Total RNA was extracted from cells and tissues with TRIzol reagent (Invitrogen) in accordance with the manufacturer's instructions. RT-qPCR was performed using an ABI 7500 Real-Time PCR System (Applied Biosystems, Foster City, CA, USA) with the SYBR Green kit (Invitrogen). β-actin was used as an internal control. Each sample was tested in triplicate. The followig primers were used for RT-qPCR analysis: ClpP, forward primer: 5′-GCC AAG CAC ACC AAA CAG A-3′, reverse primer: 5′-GGA CCA GAA CCT TGT CTA AG-3′; β-actin, forward primer: 5′-CCT GTG GCA TCC ACG AAA CT-3′, reverse primer: 5′-GAA GCA TTT GCG GTG GAC GAT- 3′.

### Western blot analysis

Total proteins were extracted with radioimmunoprecipitation assay lysis buffer (Thermo Fisher Scientific, Waltham, MA, USA). Protein concentrations were determined using the bicinchoninic acid protein assay kit (Pierce, Rockford, IL, USA). Western blot analysis

was conducted as previously described (*Li et al., 2018*) with the following antibodies: anti-ClpP (OriGene Technologies Inc., TA502075), anti-c-Src (Santa Cruz Biotechnology, sc-130124), anti-p-Src (Santa Cruz Biotechnology, sc-166860), anti-PI3K (Santa Cruz Biotechnology, sc-12930), anti-p-PI3K (Santa Cruz Biotechnology, sc-12929), anti-Akt (Wanleibio, WL0003b), anti-p-Akt (Proteintech Group, 66444-1-ig), anti-caspase 3 (Santa Cruz Biotechnology, sc-271759), anti-cleaved-caspase 9 (Wanleibio, WL01838), anti-cleaved-caspase 8 (Wanleibio, WL0153), anti-cleaved PARP (Cell Signaling Technology, #9541), anti-MMP7 (Santa Cruz Biotechnology, sc-80205), anti-E-cadherin (Abcam, ab40772), anti-vimentin (Santa Cruz Biotechnology, sc-965) and anti-β-actin (Santa Cruz Biotechnology, sc-47778), anti-mouse IgG (Cell Signaling Technology, #7076), anti-rabbit IgG (Cell Signaling Technology, #7074) and anti-goat IgG (Proteintech Group, SA00001-4).

## Immunohistochemical (IHC) analysis

All specimens were formalin-fixed, paraffin-embedded and cut into 4 μm-thick sections, which were mounted onto glass slides. IHC analysis was conducted as described previously (*Li et al., 2018*). Briefly, the slides were incubated with a primary antibody against ClpP overnight at 4 °C. Images were obtained using a Leica microscope equipped with a digital camera (Leica Microsystems, Wetzlar, Germany) and the IHC results were analyzed using Image-Pro Plus 6.0 software (Media Cybernetics, Bethesda, MA, USA). The grayscale units were converted to optical density, units and the area and integrated optical density (IOD) of the sections were then measured to calculate the mean optical density (MOD) for semi-quantitative statistical analysis, where MOD = total IOD/total area. The mean density was calculated as IOD/area.

## Cell apoptosis analysis

Cell apoptosis was detected with the Annexin V-FITC Apoptosis Detection Kit (BD Biosciences, Franklin Lakes, NJ, USA) in accordance with the manufacturer's protocol and a FACSCalibur flow cytometer (BD Biosciences).

## Colony formation assay

MDA-MB-231 and ZR-75-1 cells were seeded into triplicate wells of 6-well plates at 1,000 cells/well, and cultured for 7 days. The cells were then fixed with 4% paraformaldehyde and stained with 0.1% crystal violet solution (C0121, Beyotime Institute of Biotechnology, Haimen, China).

## Cell migration and invasion assay

Transwell chambers (8-μm pore size; Corning, NY, USA) were used to detect the migratory and invasive capabilities of BC cells. For the transwell migration assays, 200-μlL aliquots of transfected MDA-MB-231, ZR-75-1, MCF-7 and T47D cells ($4 * 10^4$ cells/well) were added into the upper transwell chamber, and 800 μL of medium containing 10% FBS were added to the lower chamber. After incubation at 37 °C under an atmosphere of 5% $CO_2$/95% air for 24 h (MDA-MB-231 and ZR-75-1 cells), or 72 h (MCF-7 and T47D cells), cells that migrated through the membrane pores were fixed in 4% paraformaldehyde for 30 min and stained with 0.1% crystal violet (DC079; Genview, Beijing, China) for 15 min at room

temperature. Matrigel$^{TM}$-coated transwell filters (Matrigel$^{TM}$: serum-free medium = 1:7, 70 μl/chamber) were used to evaluate the invasion capability of the cells. The subsequent procedures were the same as those for the cell migration assay. Cells from six random fields were counted under a microscope. All experiments were repeated three times.

## Database analysis
### TCGA dataset analyses
Gene expression data of BC tissues were downloaded from the TCGA database (https://tcga-data.nci.nih.gov/tcga/). The study cohort consisted of a total of 1010 BC patients. ClpP expression data of 112 normal breast samples were included to compare differences in ClpP expression levels in BC tissues. Overall survival (OS) and complete clinicopathological data of 990 BC patients, and RFS data of 792 BC patients were screened. ClpP expression levels were ranked from low to high based on median values. The first 50% of patients were considered as the low-expression group and the second 50% as the high-expression group.

### Kaplan Meier-plotter dataset analyses
Prognosis based on ClpP levels in BC patients was analyzed using the Kaplan Meier-plotter database (http://kmplot.com/analysis/).

### Statistical analysis
Statistical analyses were performed using IBM SPSS Statistics for Windows, version 22.0 (Armonk, NY, USA) and GraphPad Prism 7.0 software (San Diego, CA, USA). The two-tailed Student's $t$-test was used to compare two groups of independent samples. The chi-square test was used to assess the correlation between ClpP expression and the clinicopathological characteristics of BC patients. Kaplan–Meier analysis was performed to plot survival curves, which were compared with the log-rank test. A $p$-value $< 0.05$ was regarded as statistically significant.

# RESULTS
## ClpP is overexpressed in human BC tissues and associated with clinical outcomes
ClpP expression levels in BC tissues were evaluated using the TCGA dataset. The results showed that ClpP expression was significantly up-regulated in BC tissues, as compared with normal tissues ($p < 0.001$) (Fig. 1A). Receiver operating characteristic (ROC) curve analysis was conducted to estimate the diagnostic value of ClpP. The area under the ROC curve (AUC) was 0.829 (95% confidence interval [CI] [0.79–0.869], $p < 0.0001$) (Fig. 1B). To verify the results, 18 pairs of BC and adjacent normal tissues were used for RT-qPCR analysis of ClpP expression levels. Consistently, ClpP expression was significantly increased in BC tissues, as compared with normal tissues ($p < 0.01$) (Fig. 1C). The AUC was 0.713 (95% CI [0.543–0.883], $p = 0.029$) (Fig. 1D). ClpP protein expression was also assessed by IHC analysis. As compared with normal tissues, cytoplasmic ClpP immunoreactivity was markedly higher in tumor tissues ($p < 0.001$) (Figs. 1E–1G). These results showed that ClpP expression was significantly increased in BC tissues, suggesting high diagnostic potential.

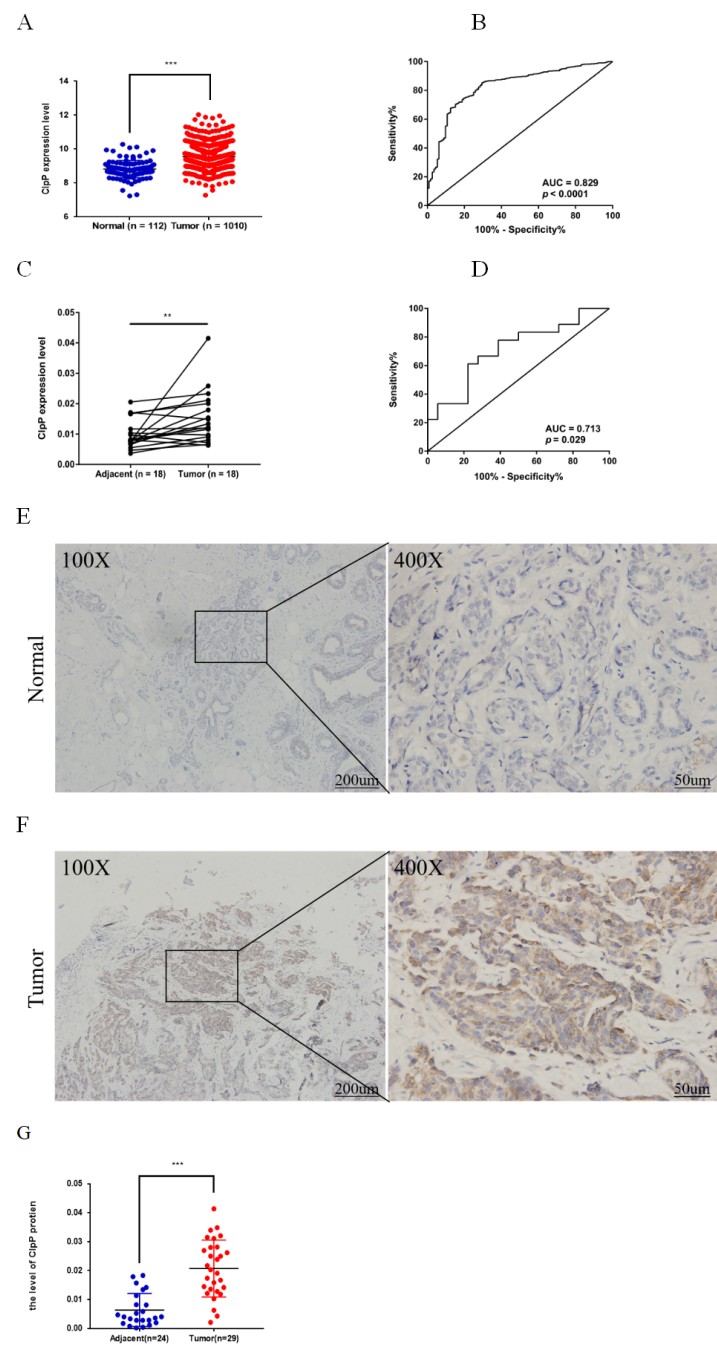

**Figure 1 ClpP is overexpressed in BC.** (A) The expression levels of ClpP were analyzed in 1010 BC tissues and 112 normal tissues in TCGA cohort. Values are presented as the mean ± SD (unpaired $t$-test). (B) ROC curve analysis of ClpP expression in TCGA cohort. (C) Comparison of ClpP expression in 18 paired BC and adjacent normal tissues by RT-qPCR. Values are presented as the mean ± SD (paired $t$-test). (D) ROC curve analysis of ClpP expression in the 18 paired tissues. (E, F) Representative IHC images of ClpP protein expression in BC and adjacent normal tissues. (G) Histogram of IHC scores of ClpP in 29 BC cases and 24 normal samples. Data are presented as the mean ± SD (unpaired $t$-test). $**p < 0.01$, $***p < 0.001$.

The clinical significance of ClpP in BC was assessed in tissues from 990 BC patients with complete clinicopathological characteristics (Table 1). High ClpP expression was found to be significantly associated with the T stage ($p = 0.0154$) and ER expression ($p = 0.0164$). Kaplan–Meier survival curves from TCGA indicated that ClpP was not associated with RFS ($p = 0.506$) or OS ($p = 0.619$) (Figs. 2A and 2B). However, within the Kaplan Meier-plotter database, high ClpP expression was associated with poor RFS ($p = 0.00071$) (Figs. 2C and 2D). These results suggest that upregulation of ClpP was correlated with the T stage, ER expression, and poor RFS, suggesting a potential essential role in BC tumorigenesis.

## Silencing of ClpP inhibits proliferation, migration, invasion and induces apoptosis of BC cells

ClpP mRNA and protein expression levels in a panel of BC and normal breast epithelial cell lines were determined. The results showed that ClpP was increased in most malignant cell lines, as compared with normal cells (Figs. 3A and 3B). To further investigate the potential biological functions of ClpP in BC, gain and loss-of-function studies were performed. The results showed that ClpP expression was significantly higher in MDA-MB-231 and ZR-75-1 cells, but lower in MCF-7 and T47D cells. Thus, ClpP was knocked-down in MDA-MB-231 and ZR-75-1 cells, and overexpressed in MCF-7 and T47D cells. ClpP mRNA and protein expression was efficiently silenced by three different siRNAs in MDA-MB-231 and ZR-75-1 cells (Figs. 3C–3F), and overexpression efficiency in MCF-7 and T47D cells was confirmed by RT-qPCR (Fig. S1A).

The suppressive effect of ClpP silencing on cancer cell growth was confirmed by the colony formation assay. The colony formation capabilities of the BC cell lines MDA-MB-231 and ZR-75-1 were markedly inhibited by si-ClpP-B ($p < 0.01$ and $<0.001$, respectively) (Figs. 4A–4E).

The transwell migration and invasion assay was performed to quantitatively assess cell metastasis and invasiveness. The results showed that cell migration and invasion were significantly reduced in cells transfected with si-ClpP-B, as compared with control cells, due to the down-regulation of MMP7 and vimentin, and the up-regulation of E-cadherin (Figs. 4F–4Q and Figs. 5O–5T). Conversely, the numbers of migratory and invasive MCF-7 and T47D cells were markedly increased by ClpP overexpression (Figs. S1B–S1E).

Flow cytometry was carried out to detect cell apoptosis. The percentages of apoptotic MDA-MB-231 and ZR-75-1 cells were increased by si-ClpP-B ($p < 0.01$ and $<0.001$, respectively), accompanied by increased levels of cleaved caspase-9, cleaved caspase-8 and cleaved poly (ADP-ribose) polymerase (PARP) (Figs. 5A–5N).

To confirm the specificity of ClpP siRNAs, the same functional experiments were performed with a second siRNA (si-ClpP-A). The results of the loss-of-function studies were consistent (Fig. S2).

These data revealed the anti-proliferative, anti-migration, anti-invasion and pro-apoptotic roles of silencing ClpP in BC cells.

**Table 1 Correlation analysis between ClpP expression levels and clinicopathological characteristics of 990 BC patients in TCGA.**

| Characteristic | Number | ClpP expression | | $\chi^2$ | p-value |
|---|---|---|---|---|---|
| | | Low (n = 495) | High (n = 495) | | |
| Age (years) | | | | 2.405 | 0.1209 |
| <55 | 406 | 215 | 191 | | |
| ≥55 | 584 | 280 | 304 | | |
| T Stage | | | | 10.4 | 0.0154[*] |
| Tx-1 | 261 | 145 | 116 | | |
| T2 | 568 | 285 | 283 | | |
| T3 | 126 | 48 | 78 | | |
| T4 | 35 | 17 | 18 | | |
| N Stage | | | | 2.431 | 0.488 |
| N0 | 486 | 234 | 252 | | |
| N1 | 330 | 168 | 162 | | |
| N2 | 103 | 58 | 45 | | |
| N3 | 71 | 35 | 36 | | |
| M Stage | | | | 1.016 | 0.3134 |
| Mx-0 | 974 | 489 | 485 | | |
| M1 | 16 | 6 | 10 | | |
| TNM Stage | | | | 2.276 | 0.517 |
| x-I | 191 | 100 | 91 | | |
| II | 560 | 276 | 284 | | |
| III | 224 | 114 | 110 | | |
| IV | 15 | 5 | 10 | | |
| ER | | | | 5.765 | 0.0164[*] |
| Negative | 232 | 100 | 132 | | |
| Positive | 758 | 395 | 363 | | |
| PR | | | | 3.84 | 0.05 |
| Negative | 327 | 149 | 178 | | |
| Positive | 663 | 346 | 317 | | |
| Her-2 | | | | 0.1181 | 0.7311 |
| Negative | 828 | 412 | 416 | | |
| Positive | 162 | 83 | 79 | | |

**Notes.**

BC, breast cancer; TCGA, The Cancer Genome Atlas; ER, estrogen receptor; PR, progesterone receptor; Her-2, human epidermal growth factor receptor-2.

*$p < 0.05$ indicates statistical significance.

## The tumor suppressive effect of silencing ClpP is mediated by the Src/PI3K/Akt signaling pathway

Previous studies have reported that the PI3K/Akt signaling pathway is involved in cellular transformation, tumorigenesis, cancer progression, and proliferation of BC cells (*Guerrero-Zotano, Mayer & Arteaga, 2016*; *Sharma et al., 2017*). Further, ClpXP reported to mediate cell migration, invasion, and metastasis *in vivo* by increasing phosphorylation of the key cellular kinases Akt and Src (*Seo et al., 2016*). Here, western blot analysis was conducted to

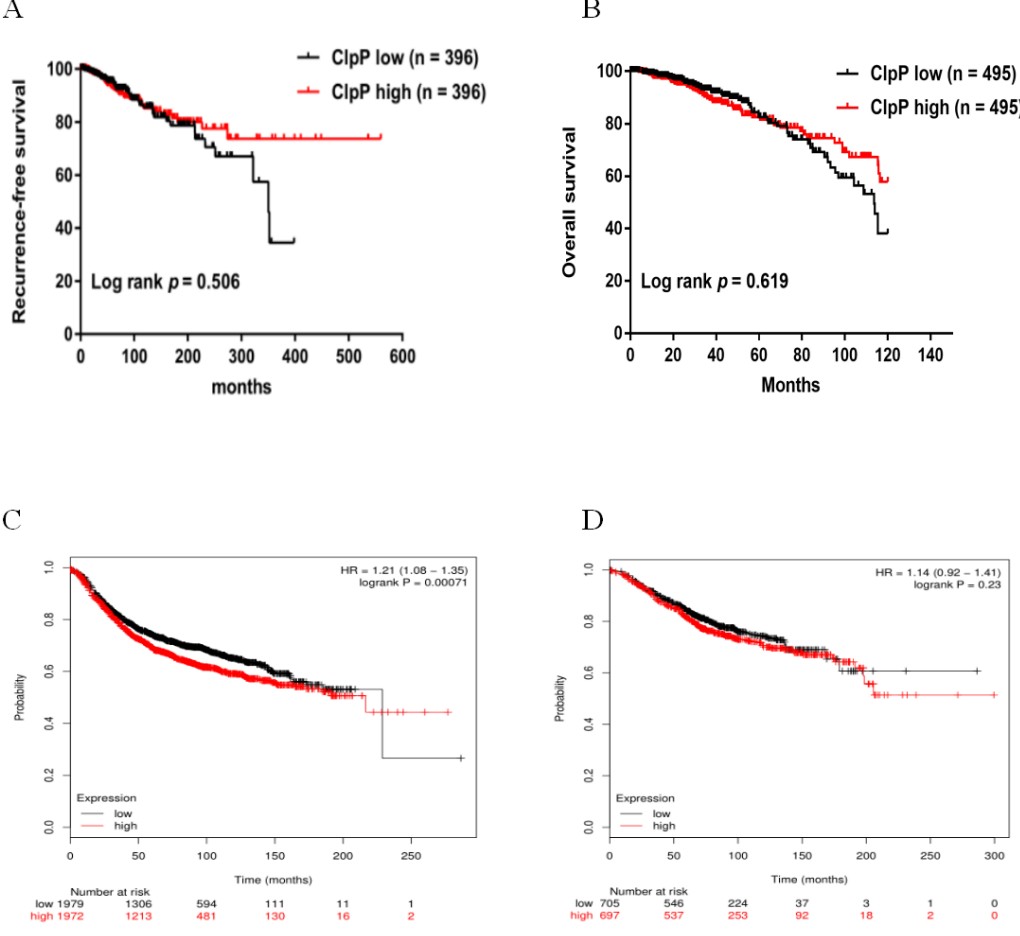

**Figure 2** **Overexpressed ClpP is associated with poor RFS.** (A, B) Kaplan-Meier analysis of RFS (A) and OS (B) in patients with BC in TCGA database. For RFS analysis, the patients were assigned to the ClpP high ($n = 396$) or ClpP low ($n = 396$) group. For OS analysis, the patients were assigned to the ClpP high ($n = 495$) or ClpP low ($n = 495$) group based on the median value of ClpP expression. (C, D) Kaplan Meier-plotter database analysis of the relationships between ClpP expression and clinical outcomes (RFS and OS). For RFS analysis, the patients were assigned to the ClpP high ($n = 1972$) or ClpP low ($n = 1979$) group. For OS analysis, the patients were assigned to the ClpP high ($n = 697$) or ClpP low ($n = 705$) group.

confirm whether the Src/PI3K/Akt pathway is involved in ClpP-induced BC progression. As shown in Fig. 6, silencing of ClpP inhibited the activation of Src, resulting in the inhibition of PI3K phosphorylation and the downstream signaling molecule Akt, which led to a series of changes in the activities of related molecules associated with cell proliferation, apoptosis, migration and invasion. These data confirmed the effect of ClpP on the proliferation and invasion capabilities of MDA-MB-231 and ZR-75-1 BC cells, as well as the induction of apoptosis.

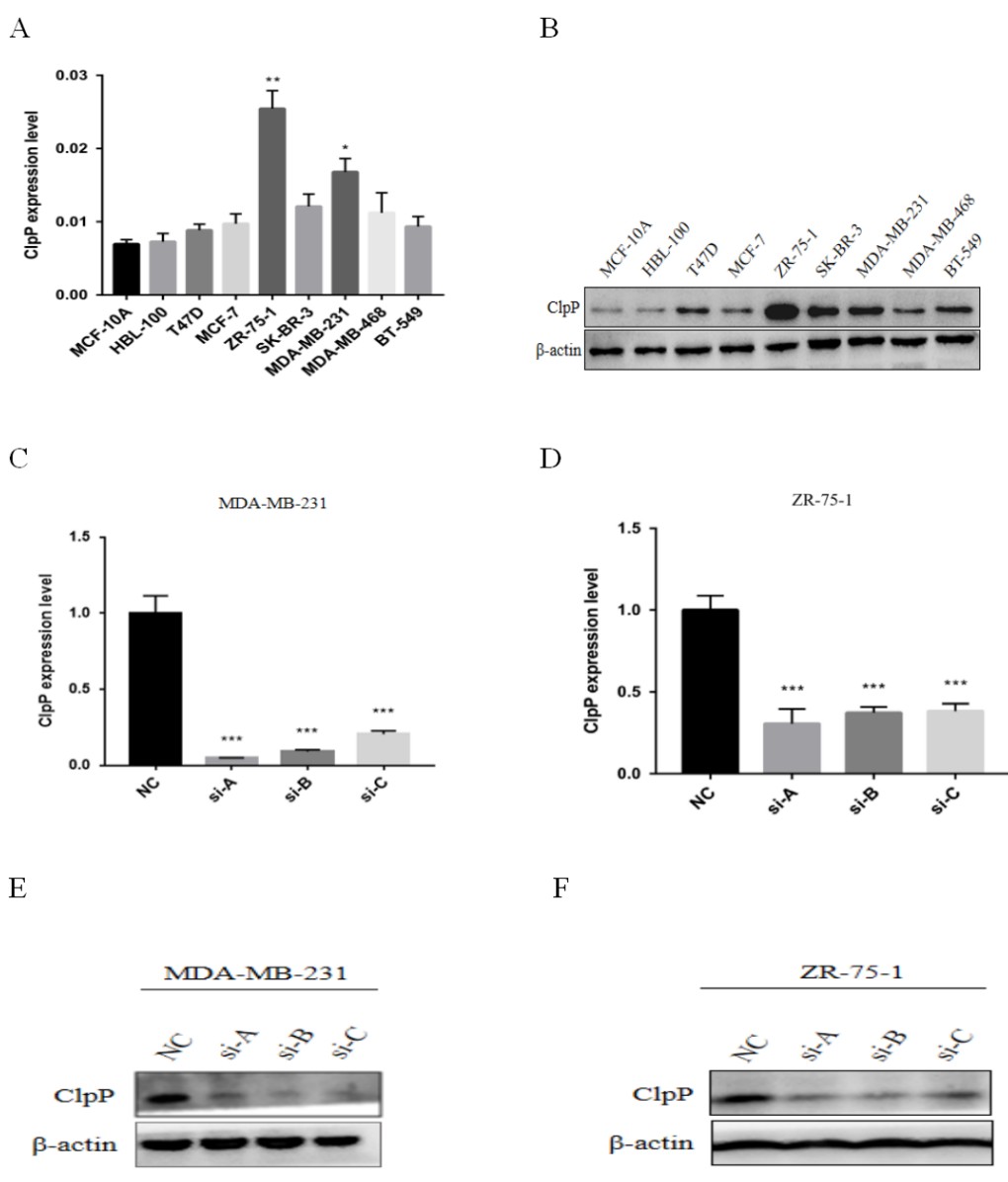

**Figure 3** **ClpP mRNA and protein expression in a panel of BC and normal breast epithelial cell lines.**
(A) RT-qPCR analysis of ClpP mRNA expression levels in BC cells, as compared to MCF-10A cells. (B)
ClpP protein expression levels in BC cells by western blot analysis, as compared to MCF-10A cells. (C, D)
ClpP expression levels were measured after transfection with three different siRNAs or negative controls
(NC) in MDA-MB-231 and ZR-75-1 cells using RT-qPCR. (E, F) ClpP expression levels were measured
after transfection with three different siRNAs or negative controls (NC) in MDA-MB-231 and ZR-75-1
cells using western blot analyses. The data are presented as the mean ± SD, *$p < 0.05$, **$p < 0.01$, ***$p <
0.001$.

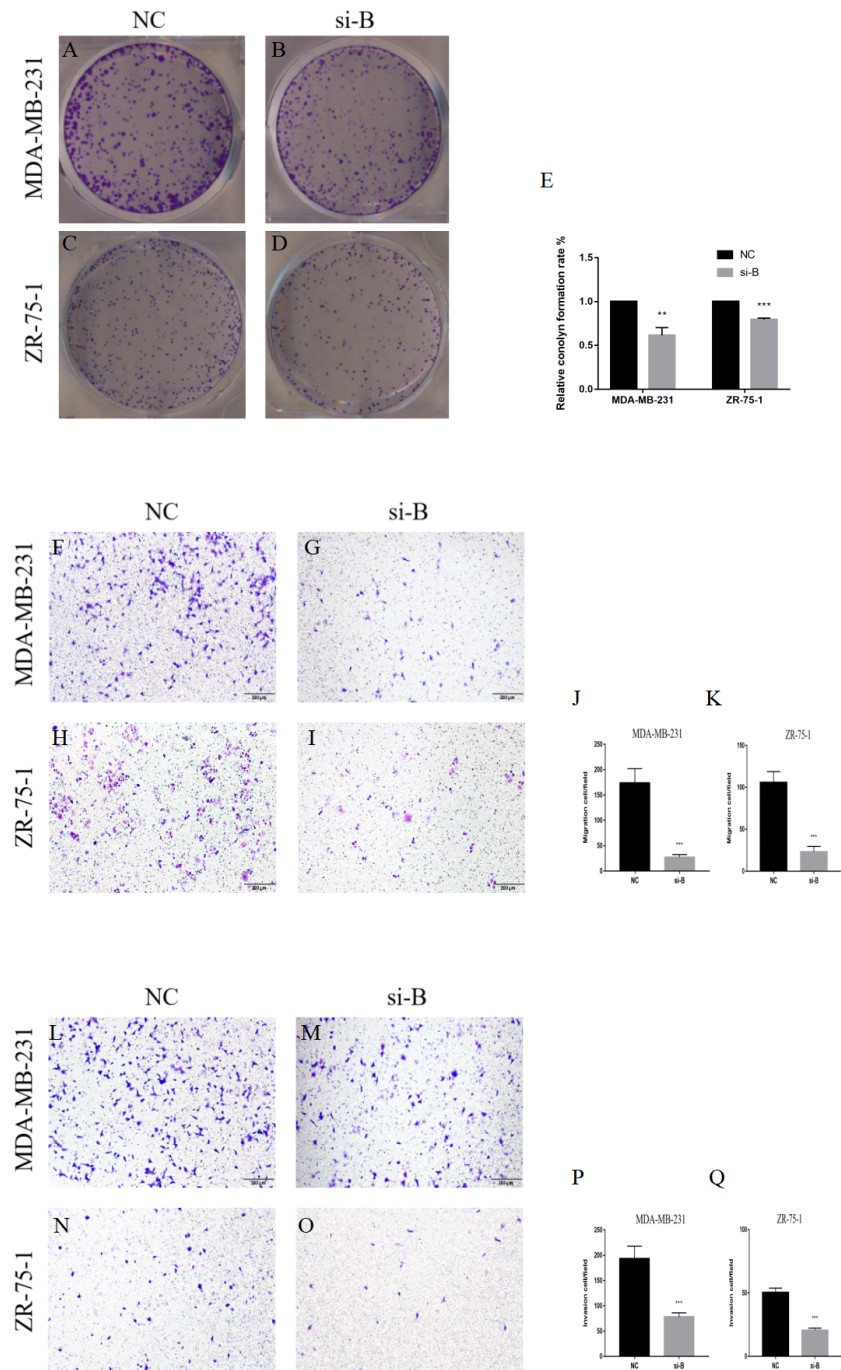

**Figure 4** **Silencing of ClpP-B inhibits proliferation, migration and invasion of BC cells.** (A–E) The effects of si-ClpP-B on BC cell proliferation were analyzed using the colony formation assay. (F–K) The effects of si-ClpP-B on BC cell migration were evaluated using the transwell assay. (L–Q) The effects of si-ClpP-B on BC cell invasion were evaluated using the transwell assay. The data are presented as the mean $\pm$ SD, $^{**}p < 0.01$, $^{***}p < 0.001$.

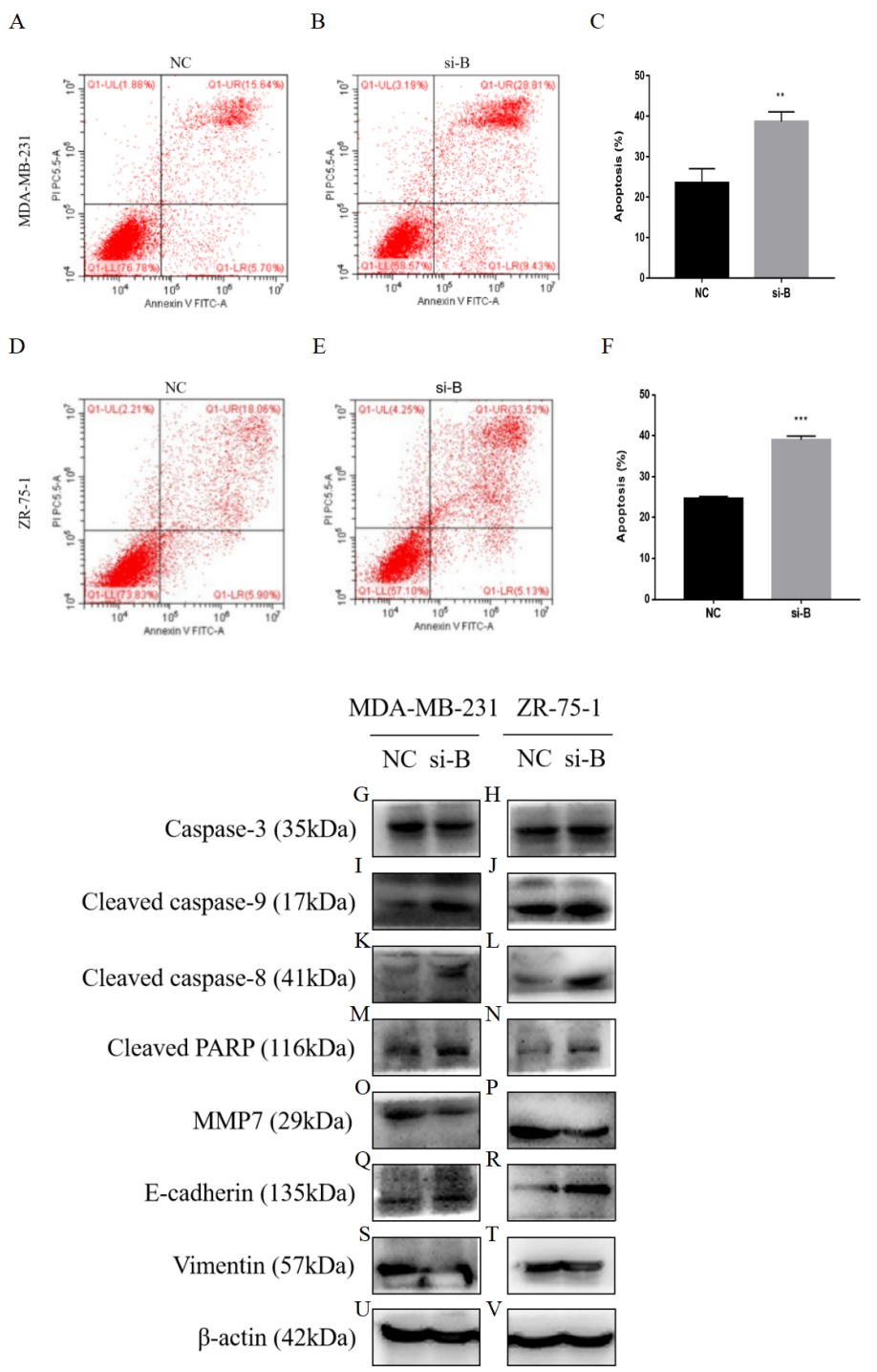

**Figure 5  Silencing of ClpP-B induces apoptosis of BC cells.** (A–C) The effect of si-ClpP-B in MDA-MB-231 apoptosis was analyzed using flow cytometry. (D–F) The effect of si-ClpP-B in ZR-75-1 apoptosis was analyzed using flow cytometry. (G–V) Western blots showing upregulation of the apoptotic markers (cleaved caspase-9, cleaved caspase-8 and cleaved PARP), the upregulated metastasis and invasiveness marker E-cadherin and down-regulated markers MMP7 and vimentin of the si-ClpP-B in MDA-MB-231 and ZR-75-1 cells. The data are presented as the mean ± SD, **$p < 0.01$, ***$p < 0.001$.

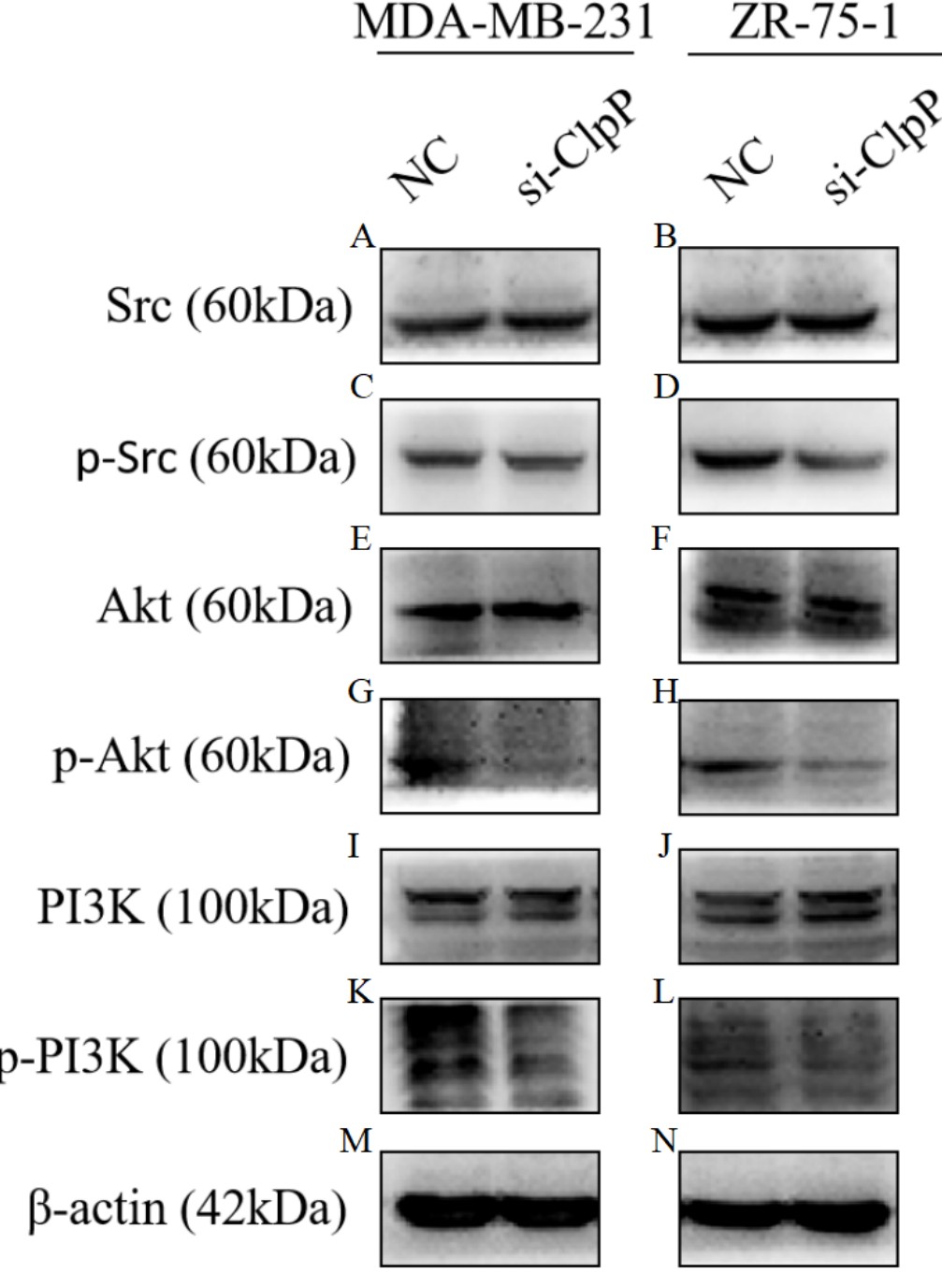

**Figure 6** **Silencing ClpP inhibits Src/PI3K/Akt signaling in MDA-MB-231 and ZR-75-1 cells.** (A, B) Western blot of Src in MDA-MB-231 and ZR-75-1. (C, D) Western blot of p-Src in MDA-MB-231 and ZR-75-1. (E, F) Western blot of Akt in MDA-MB-231 and ZR-75-1. (G, H) Western blot of p-Akt in MDA-MB-231 and ZR-75-1. (I, J) Western blot of PI3K in MDA-MB-231 and ZR-75-1. (K, L) Western blot of p-PI3K in MDA-MB-231 and ZR-75-1. (M, N) Western blot of $\beta$-actin as a control.

## DISCUSSION

A 2016 study conducted by *Seo et al. (2016)* found that ClpP was overexpressed in almost all human malignancies, as determined by immunohistochemical staining of a universal cancer tissue microarray. In the study, ClpP expression was significantly upregulated in both BC cell lines and tissues, which was consistent with previous findings. More, *Ishizawa et al. (2019)* reported that hyperactivation of ClpP could be a therapeutic strategy for patients with high ClpP expression, and that lower expression of ClpP was associated with reduced sensitivity to ClpP hyperactivation in ALM. Therefore, we can infer that high expression of ClpP presents not only a new therapeutic target for BC, but is also predictive of sensitivity to treatment. *Seo et al. (2016)* reported that ClpP expression is related to the histotype of breast adenocarcinoma. Likewise, the results of the present study showed that ClpP expression was closely correlated with the T stage and ER expression in BC, thereby demonstrating a link between ClpP expression and the clinical characteristics of BC patients.

The results of bioinformatics and a meta-analysis revealed that ClpP expression was associated with a poorer outcome in 9 (64.3%) of 14 analyzed datasets. Importantly, high ClpP expression was correlated with shorter metastasis-free survival in BC patients and reduced RFS in those with lung adenocarcinoma (*Seo et al., 2016*). Consistent with previous studies, the results of the present study showed that ClpP was not significantly correlated with OS and RFS using the TCGA dataset. However, in the Kaplan Meier-plotter database, which contains a larger sample size, high ClpP expression was associated with poor RFS in BC patients. To the best of our knowledge, the number of clinical samples determines the possible relationship between ClpP expression and survival of BC patients. The Kaplan Meier-plotter database includes data from TCGA and GEO chips, which overlap with but greatly exceed the TCGA database. Further, survival analysis of 537 French BC patients from the E-MTAB-365 database with the Kaplan Meier-plotter showed that differences in diagnostic criteria, treatment regimens, and regions will result in differences in survival benefits of the same disease, suggesting that more comprehensive information must be obtained from diverse databases in order to address these issues in future studies.

ClpP is a subunit of the ClpXP complex. *Seo et al. (2016)* found that ClpXP was associated with tumor cell migration, invasion and metastasis. Consistent with these findings, the results of the present study suggest that silencing of ClpP inhibited BC cell proliferation, migration and invasion, and induced apoptosis.

The Akt signaling pathway is important in the regulation of various cellular functions, including metabolism, growth, proliferation, survival, transcription, protein synthesis and tumorigenesis (*Aoki & Fujishita, 2017*). Mutations in the PI3K/Akt pathway can reportedly mediate development, progression and drug resistance of BC (*Guerrero-Zotano, Mayer & Arteaga, 2016*; *Sharma et al., 2017*). Src can bind to different subtypes of the integrin family, which affects the motility and metastasis of tumor cells. *Picon-Ruiz et al. (2016)* found that the invasion of localized fat by cancer cells will activate Src, maintain the production of pro-inflammatory cytokines, and promote metastasis of BC cells. The results of the present study demonstrated that phosphorylation of Src, PI3K and Akt was markedly

downregulated by silencing ClpP. These findings provide powerful evidence that silencing ClpP contributes to inactivation of the Src/PI3K/Akt pathway in BC.

In summary, the present study provides the first evidence that ClpP is frequently up-regulated in BC and that ClpP has high diagnostic value.

## CONCLUSIONS

ClpP expression is markedly increased in BC and significantly correlated with the T stage, ER expression, and poor RFS. Silencing of ClpP significantly inhibited proliferation, migration and invasion, and promoted apoptosis of BC cells, resulting in suppression of the Src/PI3K/Akt signaling pathway. These data indicate that ClpP is an oncogene, and may be a promising diagnostic biomarker and therapeutic target in BC.

## ACKNOWLEDGEMENTS

We thank our tutors and friends for their help and encouragement.

### Funding

The authors received no funding for this work.

### Competing Interests

The authors declare there are no competing interests.

### Author Contributions

- Juan Luo conceived and designed the experiments, performed the experiments, analyzed the data, prepared figures and/or tables, authored or reviewed drafts of the paper, and approved the final draft.
- Beilei Zeng and Guosheng Ren conceived and designed the experiments, authored or reviewed drafts of the paper, and approved the final draft.
- Chunfang Tao performed the experiments, prepared figures and/or tables, and approved the final draft.
- Mengqi Lu analyzed the data, prepared figures and/or tables, and approved the final draft.

### Human Ethics

The following information was supplied relating to ethical approvals (i.e., approving body and any reference numbers):

The Institutional Ethics Committees of the First Affiliated Hospital of Chongqing Medical University approved this study (approval no. 2019-208).

### Data Availability

Data are available in the Supplemental Files.

## Supplemental Information

Supplemental information for this article can be found online at http://dx.doi.org/10.7717/peerj.8754#supplemental-information.

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
