# Peer review of "ClpP regulates breast cancer cell proliferation, invasion and apoptosis by modulating the Src/PI3K/Akt signaling pathway"

_PeerJ, doi:10.7717/peerj.8754_

## Round 0.1 · original submission · Major Revisions

As the reviewers recommend, it would help if professional help is sought for fixing grammatical mistakes and syntax errors. Address all the concerns raised by both the reviewers with special attention to the discrepancy in TCGA and Kaplan-Meier data and additional experiments.

Reviewer 1 ·

Basic reporting

1. English needs to be proof-read by native speaker or professional proof-reader
2. Introduction needs substantial modification to provide more background
3. Figures need to be properly labelled, few figures need to be replaced with better quality images (refer to general comments)

Experimental design

No comment

Validity of the findings

Refer to general comments

Additional comments

Figure 1
• Please provide high quality representative images for IHC data in Fig. 1E. It is hard to draw any conclusion based on the image provided in this manuscript.
• Fig. 1F represents quantification of IHC data thus figure legend and result section should be appropriately modified.

Figure 2
• Authors observed discrepancy in results from TCGA and Kaplan-Meier Plotter database for RFS and OS. Please comment on it in discussion section.

Figure 4
• Quantification of colony formation assay data does not correlate with the actual effect observed in provided images. Please provide images from all the replicates in supplementary. Also, number of cells in control well for ZR-75-1 seems to be less than that for MDA-MB-231.
• Invasion assay for ZR-75-1 should be repeated.

Figure 5
• Figure legend is missing
• Western blot result for p-PI3K is not very clear. Please provide better quality image to support any conclusion drawn

Reviewer 2 ·

Basic reporting

Generally, I found the manuscript to be written clearly. However, there were occasional lapses and mistakes that could have been avoided if the authors have had the manuscript read and edited by someone whose native language is English. E.g. In their abstract (line 20), they written “Case in hydrolytic protease (ClpP)…”, clearly it was meant to be “Casein in hydrolytic protease (ClpP)…” Another example early on in the manuscript would be at line 47, “…support cell growthy by different mechanisms to the surround tissues.” This sentence does not make any sense and I attribute it to a language error. I would therefore suggest the authors enlist the help of someone in future revisions of the manuscript to avoid such mistakes.

Recently, there was a published study in Cancer Cell on ClpP, see (https://doi.org/10.1016/j.ccell.2019.03.014). It would be good if the authors can cite and discuss the study in their manuscript and perhaps discuss their findings in light of this recent study.

*Note* The title of Figure 4 in their submitted manuscript is wrong (page 23 of submitted manuscript). It was titled, “Correlation analysis between ClpP expression levels…” I believed that is mistake as that is the title of their Table 1 (page 29 of manuscript).

Experimental design

With regards to their clinical results, I find that the data that ClpP is over-expressed in breast cancer to be convincing. However, I do not find the prognostic value of ClpP to be convincing. Firstly, the authors did not discuss the discrepancies of their findings from TCGA data and KM-plotter. Furthermore, the prognostic value of ClpP seems to be modest. There might be other clinical variables that affect the prognostic value of ClpP. The authors should therefore look deep into that to perhaps identify other co-variates influencing survival.

Concerning their functional studies, I would suggest the authors perform some gain-of-function studies, given that ClpP was identified to be over-expressed in breast cancer. Furthermore, in their loss-of-function studies, they only demonstrated results using 1 si-RNA while they had showed 3 effective siRNAs that can knockdown ClpP levels. It would be good for them to demonstrate the same effect with a 2nd siRNA and also perform rescue experiments to show specificity of ClpP to the phenotypes in questions.

Validity of the findings

Finally, I think that the authors did not clearly discuss their hypothesis or how they linked ClpP to the Akt signaling pathway. Was there any evidence to suggest Akt has a functional relationship to ClpP? If so, I seem to be missing the discussion in the manuscript. It seems to me that the authors just jump straight into looking at Akt signaling without demonstrating a relationship between ClpP and Akt. Is there protein-protein interaction or gene expression regulation? I would want the authors to further clarify and present data discussing the relationship.

Additional comments

No other comments beyond what is mentioned above.

---

## Round 0.2 · accepted · Accept

Dear Dr. Ren,

Our reviewers have gone through the revision and are satisfied with the rebuttal. Your manuscript has been now accepted. Congratulations.

Reviewer 1 ·

Basic reporting

No comment

Experimental design

No comment

Validity of the findings

No comment

Additional comments

Authors have addressed all my concerns to satisfaction

Reviewer 2 ·

Basic reporting

No comments

Experimental design

No comments

Validity of the findings

No comments

Additional comments

The authors have tried their best to address my concerns within the time constraints. Further investigations would be outside the scope of the current manuscript.